# Synthesis of TiO_2_/Al_2_O_3_ Double-Layer Inverse Opal by Thermal and Plasma-Assisted Atomic Layer Deposition for Photocatalytic Applications

**DOI:** 10.3390/nano13081314

**Published:** 2023-04-08

**Authors:** Hamsasew Hankebo Lemago, Feras Shugaa Addin, Dániel Atilla Kárajz, Tamás Igricz, Bence Parditka, Zoltán Erdélyi, Dóra Hessz, Imre Miklós Szilágyi

**Affiliations:** 1Department of Inorganic and Analytical Chemistry, Faculty of Chemical Technology and Biotechnology, Budapest University of Technology and Economics, Műegyetem rkp. 3., H-1111 Budapest, Hungary; 2Department of Organic Chemistry and Technology, Faculty of Chemical Technology and Biotechnology, Budapest University of Technology and Economics, Műegyetem rkp. 3., H-1111 Budapest, Hungary; 3Department of Solid-State Physics, Faculty of Sciences and Technology, University of Debrecen, H-4002 Debrecen, Hungary; 4Department of Physical Chemistry and Materials Science, Faculty of Chemical and Bioengineering, Budapest University of Technology and Economics, Műegyetem rkp. 3., H-1111 Budapest, Hungary; 5MTA-BME Lendület Quantum Chemistry Research Group, Műegyetem rkp. 3., H-1111 Budapest, Hungary

**Keywords:** inverse opal, TiO_2_, Al_2_O_3_, nanocomposites, thermal ALD, plasma ALD, photocatalysis

## Abstract

In comparison to conventional nano-infiltration approaches, the atomic layer deposition (ALD) technology exhibits greater potential in the fabrication of inverse opals (IOs) for photocatalysts. In this study, TiO_2_ IO and ultra-thin films of Al_2_O_3_ on IO were successfully deposited using thermal or plasma-assisted ALD and vertical layer deposition from a polystyrene (PS) opal template. SEM/EDX, XRD, Raman, TG/DTG/DTA-MS, PL spectroscopy, and UV Vis spectroscopy were used for the characterization of the nanocomposites. The results showed that the highly ordered opal crystal microstructure had a face-centered cubic (FCC) orientation. The proposed annealing temperature efficiently removed the template, leaving the anatase phase IO, which provided a small contraction in the spheres. In comparison to TiO_2_/Al_2_O_3_ plasma ALD, TiO_2_/Al_2_O_3_ thermal ALD has a better interfacial charge interaction of photoexcited electron–hole pairs in the valence band hole to restrain recombination, resulting in a broad spectrum with a peak in the green region. This was demonstrated by PL. Strong absorption bands were also found in the UV regions, including increased absorption due to slow photons and a narrow optical band gap in the visible region. The results from the photocatalytic activity of the samples show decolorization rates of 35.4%, 24.7%, and 14.8%, for TiO_2_, TiO_2_/Al_2_O_3_ thermal, and TiO_2_/Al_2_O_3_ plasma IO ALD samples, respectively. Our results showed that ultra-thin amorphous ALD-grown Al_2_O_3_ layers have considerable photocatalytic activity. The Al_2_O_3_ thin film grown by thermal ALD has a more ordered structure compared to the one prepared by plasma ALD, which explains its higher photocatalytic activity. The declined photocatalytic activity of the combined layers was observed due to the reduced electron tunneling effect resulting from the thinness of Al_2_O_3_.

## 1. Introduction

The rapid growth of the population and industrial development resulted in the discharge of toxic pollutants into the environment and the ecosystem as well [1]. Although the concentrations of the contaminants, such as pharmaceuticals, insecticides, surfactants, and endocrine disruptors (i.e., hormones), are found in minimal amounts, a profound effect on the environment has been observed [2,3,4]. In recent years, different approaches have been developed to reduce the harmful effect of trace organic chemicals from natural waters. Promising results and progress for semiconductor oxide inverse opals (IOs) in water purification processes have been frequently reported [5,6]. IOs are photonic crystals of a highly ordered three-dimensionally interconnected microporous skeleton accompanied by a high specific surface area and unique optical characteristics, such as the optical band gap [7], photo localization [8,9], negative refractive index [10], slow light effect [11,12], and super prism effect [13], for various applications such as photocatalysis [14,15], gas sensors [16], and energy storage [17,18]. These initiatives may boost the photocatalytic activity by increasing the electron–hole pair separation efficiency in the IO structures and extending the photo response range [19,20]. The growth of composites is a viable strategy to enhance photocatalytic performance and a useful means to accelerate the separation of electron–hole pairs in IO structures [21,22].

In fabricating the IO structures, a bottom-up approach is the most popular synthetic technique. This process passes through three different successive steps: (a) the deposition of a colloid crystal template on top of the substrate; (b) filling the nanosphere’s voids with the precursors of high-dielectric materials [23,24]; and (c) removing the original opal template using wet chemical etching and calcination in the case of SiO_2_ and polymer templates, respectively [25,26]. Additionally, techniques such as atomic layer deposition (ALD) [27,28], sol–gel [29,30], and hydrothermal methods [31,32] could be used for infiltration purposes. However, due to the extremely small void area between nanospheres, conventional infiltration processes, including chemical vapor deposition (CVD) and sol–gel, have limitations in completely infiltrating oxides in the voids of the template [33]. Since the aforementioned approaches have limitations in the infiltration of the voids, approaches such as ALD have been recently proposed to reverse these and associated effects.

Atomic layer deposition can be used to develop semiconductor oxides, such as TiO_2_, ZnO, WO_3,_ Al_2_O_3_, and multiwalled nanostructures (i.e.,TiO_2_/ZnO, TiO_2_/FeO_3_, and Al_2_O_3_/TiO_2_) [34,35,36]. It allows atomic-scale control over thickness and a homogeneity vapor-phase thin-film growth technique. It is based on a succession of two self-limiting interactions between gas-phase precursor molecules and a solid surface. The self-limited surface reactions ensure that the films deposited by ALD grow atomically layer-by-layer, allowing accurate thickness control and the conformal deposition of thin films on high-aspect-ratio nanostructures [19].

Due to its potency in decomposing various pollutants into biodegradable compounds, TiO_2_ has been found an effective photocatalyst for the breakdown of pollutants [37,38,39] and it is also a widespread material to fabricate IOs. Moreover, recent studies have proven that the development of double-layered IOs with enhanced photocatalytic activity, such as TiO_2_/ZnO [40], TiO_2_/Fe_3_O_4_ [41], ZrO_2_/TiO_2_ [42], and Cu-doped TiO_2_ [43], have been shown to accelerate charge separation efficiency and extend the photoexcitation energy for efficient photocatalysis. Given that coupling effects may be attributed to the synergistic interfacial charge interactions among composites, this resulted in more interface between layers, for the efficient transfer of photo-generated electron–hole pairs to restrict the recombination. In addition, heterojunction at the oxide interfaces facilitates accelerated charge separation efficiency and the transfer of photoinduced charge carriers more than single bulk oxides [44,45,46].

Aluminium dioxide (Al_2_O_3_), however, is not considered to have photocatalytic activity, since it is usually amorphous with several defect sites, which can be recombination centers [47]. Nonetheless, the ALD-grown ultrathin amorphous Al_2_O_3_ layers exhibited photocatalytic properties. It was a question of how an ALD-grown TiO_2_/amorphous Al_2_O_3_ heterojunction would work as a photocatalyst. It has been seen that an ultra-thin-film Al_2_O_3_ ALD suppressed the photocatalytic activity of the TiO_2_ nano-particles [48,49], whereas E. Barajas-Ledesma et al. found that the Al_2_O_3_ doping enhances the photocatalytic activity of TiO_2_ thin films. Shenouda et. al. [50] also revealed that the ultra-thin-film Al_2_O_3_ ALD is an active photocatalyst for wastewater treatment. In this case, it was unclear whether the Al_2_O_3_ layer would also have a photocatalytic effect. The influence of thermal or plasma ALD for preparing photocatalytic Al_2_O_3_ had to be explored. 

Hence, this study followed a two-step complete growth process to prepare TiO_2_, TiO_2_/thermal Al_2_O_3,_ and TiO_2_/plasma Al_2_O_3_ inverse opal structures. Initially, TiO_2_ was grown on the PS opal by thermal ALD and the template was removed at a higher temperature to obtain the IO structure. Secondly, a thin layer of Al_2_O_3_ was also grown on the IO surface by using thermal or plasma ALD, and TiO_2_/thermal Al_2_O_3_ and TiO_2_/plasma Al_2_O_3_ plasma double layers were obtained. The opal template was prepared by vertical layer deposition using a glass substrate. This study aimed to fabricate TiO_2_ single and TiO_2_/Al_2_O_3_ double-layered IO composite structures and explore their photocatalytic activities. In order to determine the proper annealing temperature, thermal analysis was employed. The bare IO and its composite structures were studied by SEM-EDX (scanning electron microscope–energy dispersive X-ray microanalysis), XRD (X-ray diffraction), Raman, and UV–Vis spectroscopy. 

## 2. Material and Methods 

### 2.1. Preparation of Inverse Opal

The opal template was prepared from a PS suspension (Sigma Aldrich, Darmstadt, Germany,10% *w*/*w* and 600 nm particle size) after dilution to achieve a concentration of 0.3% by transferring 0.15 mL to a 5 mL flask and them placed in a closed glass container to avert its evaporation and ultrasonicated for 2 h to separate the aggregated nanospheres. A microscope glass slide (Knittel Glaser ca. 76 mm × 26 mm, Braunschweig, Germany) was used as a planar substrate, which was cleaned first with soap, ethyl alcohol, and ion exchange water. This substrate was soaked in a piranha solution (a mixture of H_2_SO_4_ and H_2_O_2_ in a 3:1 ratio) for 1 h to remove any organic residue on the glass surface and to make it more hydrophilic. The clean glass slides were placed in the PS suspension at a 45-degree angle position to achieve the successful vertical deposition self-assembly of colloidal particles. They were placed in the furnace at 50 °C for 14 h, so the colloidal crystal could form during the evaporation of the water through colloidal self-assembly, and were finally heated to 80 °C for 1.5 h. The schematic representation of the whole experimental procedure is shown in Figure 1.

The ALD parameters for TiO_2_ and Al_2_O_3_ deposition are shown in Table 1. For the ALD runs, a Beneq TFS-200-186 ALD reactor was used, while for TiO_2_ and Al_2_O_3_ depositions, TiCl_4_, TMA, and H_2_O precursors were used at 52.9 °C in a thermal-mode ALD. For plasma-assisted Al_2_O_3_ ALD deposition, TMA and O_2_ gas precursors were also used. Prior to the sample preparation, the deposition chamber had been heated to 45–50 °C. The pressure inside the vacuum chamber was 7.4 m bar while it was 1.2 m bar in the reactor. The RF power used for the plasma is 50 W. 

Spectrometry ellipsometry was used to determine the film thickness of the samples. Si wafers were also introduced into the ALD system with the samples together to act as a seed layer and reference layer for the oxides.

Finally, the PS opal template was removed by annealing in a Nabertherm L9/11/B410 furnace. The samples were annealed in the air heated from room temperature to 500 °C for 4 h, and held there for 2 h, leaving behind a mechanically stable inverse opal.

### 2.2. Sample Characterization 

For the PS opal sample, thermal analysis was carried out with a TA Instruments SDT 2960 simultaneous thermogravimetry/differential thermal analysis/derivative thermogravimetry (TG/DTG/DTA) device, in an air atmosphere with a heating rate of 10 °C/min and a flow rate of 130 cm^3^/min until 600 °C, simulating the environment during the removal of the polystyrene nanospheres. The evolved gaseous products were identified with a thermostat GSD 300 Balzers Instruments type quadrupole mass spectrometer (MS) coupled online to the TG/DTA device. The selected ions were followed in multiple ion detector (MID) modes. For the measurement, the thin PS film was scratched off from the glass surface and collected into the Pt crucibles. 

The SEM images were taken with an FEI Inspect S50 scanning electron microscope, in a low vacuum system, using a backscattered electron detector. The samples were examined without any modification or treatment, and no conductive film (Au and Pd) was deposited on the samples to avoid possible signals due to their interference. 

EDX spectra were measured by a Bruker Quantax 200 SEM at 20 kV. The elements were identified, and their atomic percentages in the study spots were calculated. 

Raman spectra were measured by a Jobin Yvon Labram Raman spectroscope hyphenated with an Olympus BX41 microscope, using a green (532 nm) Nd-YAG laser in a range of 100–1800 cm^−1^.

X-ray diffractograms were recorded on a PAN analytical X’Pert Pro MPD X-ray diffractometer using Cu Kα radiation; the measurement range was 5°–65°. Diffraction patterns were referenced against the ICDD database for sample identification.

The photoluminescence spectra (PL) were recorded at room temperature on an Edinburgh Instruments FS5 spectrofluorometer. The excitation wavelength was 380 nm, and a 400 nm notch filter (FGL400) was placed in the emission beam. 

A variable angle spectroscopic ellipsometer (SE, SEMILAB SE-2000) was used to determine the film thickness of the samples.

The reflectance UV–Vis spectra were recorded by the Avantes AvaSpec-2048 spectrophotometer. 

The photocatalytic activity of the thin films was tested by the degradation of methylene blue (MB) dye under visible light illumination for 3 h. The test was carried out by immersing the film in a glass container with a 20 mL of 1.0 × 10^−6^ M MB solution. For visible light irradiation, six 18 W fluorescent lamps were placed in a stack of three lamps in two lines 10 cm apart. The sample solution was located 5 cm in front of the middle of the two lamps. The solution was thoroughly mixed by a magnetic stirrer for half an hour to allow adsorption–desorption equilibrium to occur. The decomposition of the dye was measured by an Avantes AvaSpec-2048 fiber optic spectrometer with an Ava Light–DHS light source and evaluated by its Ava Soft software. In every half-hour, 3 mL was taken from the solution, whose absorbance was measured in a quartz cuvette and was put back for further photocatalysis. The decomposition of the dye was examined by the decrease in the relative absorbance (A_t_/A_0_) at the most intensive peak of MB at 665 nm. 

## 3. Result and Discussion

### 3.1. Opal Template as Determined by Thermal Analysis

The thermal analysis of the PS opal template showed that annealing at 600 °C was an effective technique to remove the template (Figure 2). It was observed that the template was thermally stable up to about 100 °C, with only an evolution of solvent (mass to charge ratio, *m*/*z* = 18, 4.6% mass loss). The TG/DTG curves showed two main stages (300–425 and 425–500 °C) of decomposition combined with the exothermic burning of the organic material. The first step was explained by the decomposition and combustion of the main part of the sacrificial polymer precursors, while the second step was the result of the burning of the remaining char. Based on the measurement, 500 °C was adequate as an annealing temperature to remove the polymer opal template and produce the IO structure. This also corroborated to Worzakowska [51] and Manikandan et al. [52]. 

### 3.2. Morphological and Compositional Evaluation of the Samples

TiO_2_ and Al_2_O_3_/TiO_2_ IOs arrays were analyzed through infiltration methods in which the filling materials entered the PS nanosphere microstructure. The SEM images of an opal showed that they are highly porous and have uniform sphere sizes as shown in Figure 3. The IO and double-layer samples no longer showed consistent differences. In the template structure perspective, the layer’s interconnected and well-ordered periodic microporous structures also displayed a face-centered cubic (FCC) formation. However, minor cracks were observed due to the substantial volume shrinkage during annealing, as the IO shells contract by 16.67% after annealing. 

As shown in Table 2, which details the composition of the samples from the EDX analysis, the pure IO and double-layer samples contained considerable amounts of Ti, Al, and O. Additionally, Cl and C were residues of TiCl4 and TMA precursors, and signals from the glass substrate (Na, Al, Si, and Ca) were detected.

### 3.3. XRD and Raman Structural Analysis 

The crystalline structure of the samples was characterized by both XRD and Raman (Figure 4a,b). The major XRD peaks were shown at 25.6°, 38.2°, 48.6°, 55.4°, and 63.1°, corresponding to TiO_2_ tetragonal anatase phase (ICCD card number: 98-009-2363). Similar results were reported by Yew et al. [53]. Hence, the annealing temperature (500 °C) was sufficient to crystallize the amorphous TiO_2_ to a stable anatase phase, which corroborated Chakraborty et al. [54]. Due to their trace amounts, the 5 nm Al_2_O_3_ layers synthesized by thermal or plasma ALD on IO were not enough to yield the XRD signal in the analysis, especially as the as-grown Al_2_O_3_ were expected to be amorphous. The crystal structure of the IO has not been altered by the thin Al_2_O_3_ layer grown by thermal ALD. In contrast to the thermal ALD, after the plasma reaction, peaks at 25.6°, 38.2°, and 48.6° were observed only, which resulted in a decrease in the peak intensity and crystallinity of the double layer. In addition, the broad diffraction band with no peaks between 20° and 30° is a typical soda lime glass used as a substrate.

The results from Raman spectroscopy (Figure 4b) showed peaks at ~142, ~395, ~519, and ~637 cm^−1^, which corresponded to tetragonal anatase TiO_2_ IO. Scepanovic et al. [55] reported a similar trend of Raman shifts for TiO_2_ IO nanostructure. Similar to XRD, the thin layers of amorphous Al_2_O_3_ grown by thermal or plasma ALD were not detected. Nevertheless, the polymer templates were removed entirely with annealing from the structure, as no D and G peaks of the carbon were visible. 

### 3.4. Optical Analysis 

The pure TiO_2_ and its nanocomposite IO samples showed a sharply increased absorption intensity at shorter wavelengths with absorption edges of 330, 320, and 335 nm for TiO_2_, thermal, and plasma Al_2_O_3_ ALD-grown double-layer nanocomposites, respectively, as indicated in Figure 5a–c. Therefore, these strong edges of absorption amplified the extent of the light absorption due to the slow photon effects, i.e., light propagation at reduced group velocity, [56,57], evaluation through electronic transitions, and generating electron–hole pairs necessary for photocatalytic processes. The composite samples exhibited absorption edges in the visible region corresponding to 470, 630, and 800 nm of the thermal and 490, 610, 672, and 800 nm, of the plasma ALD samples, respectively. These sinusoidal absorption patterns in the vis region were comparable to the layer thickness and also the presence of Al_2_O_3_ ALD ultra-thin films in combination with an energy band gap (3.6 eV and 3.8 eV). 

The basic absorption energy, which corresponds to the electron excitation from the valence band to the conduction band, was used to calculate the energy band gap (E_g_) value of the thin films. The Tauc model was used for the absorption spectrum fitting approach to estimate the E_g_ of the sample [58].
(αhν)^1/n^ = A (hν − E_g_) (1)
where α is the absorption coefficient, E_g_ is the energy band gap, hν is the photon energy, and A is the constant exponent analogous to the type of transition that occurs in the material where n = ½, and when it equals 2 it is corresponding to the allowed indirect and direct transition states. 

In this study, the Tauc plot was calculated by n = ½ for the indirect band gap. This was achieved by plotting the graph between (ahv)^1/n^ on the *Y*-axis and photon energy (E_g_) on the *X*-axis and extrapolating the linear portion of the graph onto the *X*-axis of the linear region of the plot to (αhν)^1/n^ = 0. 

The evaluated indirect optical band gap values of the TiO_2_, TiO_2_/Al_2_O_3_ thermal, and TiO_2_/Al_2_O_3_ plasma ALD IO samples were found to be 3.2, 3.6, and 3.8 eV, respectively. In fact, the bulk Al_2_O_3_ materials have a higher band gap value, 8.7 eV [59], and thus the composite TiO_2_/Al_2_O_3_ was shifted the band gap to a reduced value. Due to the highly amorphous and less-ordered structure of the plasma ALD-grown Al_2_O_3_, the determined band gap value was lower than those reported for the plasma ALD Al_2_O_3_ layer thickness of 8 nm (4.3 eV) and 38 nm (4.75 eV) [39]. Hence, these results are in agreement with the literature for ultra-thin-film Al_2_O_3_, as reported by Shi et.al. [60] and Costina [61]. 

### 3.5. Photoluminescence (PL) Studies 

Figure 6 shows the PL spectra of the three photocatalyst samples, obtained at room temperature, using an excitation wavelength of 380 nm. With an initial threshold wavelength of 420 nm and a cut-off wavelength of 800 nm, the shapes of the three curves are similar. Their broad peaks are located at close positions between ~515 nm (3.86 eV) and ~540 nm (3.68 eV). The wavelengths of the PL bands of our samples are close to the wavelength of the ‘green band’ observed around 515 nm in the PL spectra of anatase TiO_2_ films grown by ALD at different temperatures, where a ‘red band’ also appeared around 600 nm [62]. (The ‘green band’ was correlated with the surface oxygen vacancies on anatase TiO_2_ films; the ‘red band’ was related to defects of under-coordinated Ti^3+^ ions.) The relative intensities of the PL bands of our samples seem to correlate with the degree of their crystallinity. The TiO_2_/Al_2_O_3_ thermal film shows the most intense PL, which exhibits the strongest (101) anatase peak in the XRD pattern. The TiO_2_/Al_2_O_3_ plasma film produces the weakest luminescence of the three samples and this sample has only a weak (101) anatase peak in its powder diffractogram. In brief, the higher crystallinity shown by the XRD pattern is accompanied by the lower concentration of the non-radiative recombination centers, which is indicated by the more intense PL.

### 3.6. Photocatalytic Studies

The results of the photocatalytic activity of the samples are shown in Figure 7 and Figure 8. These figures further show the decolorization of MB dye in the solution of TiO_2_ IO and Al_2_O_3_/TiO_2_ thermal and plasma ALD under UV and visible light irradiation as a function of time (Figureand the decolorization of the MB in the solution, pseudo-defined by Equation (2),
Degradation (%) = (A_t_/A_0_) × 100% (2)
where A_t_ = the absorbance of the sample solution at time t, and A_0_ = the initial concentration. Based on the Beer–Lambert law, the A_t_/A_0_ ratio was taken equal to the absorbance ratio A_t_/A_0_ with A_t_ and A_0_ being the absorbance at the 665 nm band maximum of MB (after subtracting the background) at time t and at t = 0, respectively.

The rate of the decolorization of the MB in the sample solution and the rate constant were evaluated by using a pseudo-first-order equation, Equation (3)
ln [C_0_/C_t_] = Kt(3)
where C_0_ is the initial concentration of MB, C_t_ is the concentration after irradiation time t, and K is the rate constant.

The results of the present study were almost in agreement with the linear equation, and it was discovered that the photolysis reactions of the MB in the samples obeyed a pseudo-first-order kinetic model shown in the literature [63,64]. The linear plots of ln (C_0_/C_t_) versus irradiation time, t, are shown in Figure 8b. The rate constant k values were calculated from the slopes of the lines and the coefficient of linear regression, R was calculated and summarized in the inset of Table 3. The summary makes it evident that the rate constant of the bare TiO_2_ has the largest k value, while both the combined thermal and plasma ultra-thin-film ALD Al_2_O_3_ layers exhibited decreased k values.

The measurement showed that all samples were effective in the decolorization of MB under visible light illumination, as shown in Figure 8a. The TiO_2_ IO achieved faster decolorization with a rate of 35.4% of MB in 3 h, while the composites, thermal and plasma Al_2_O_3,_ had MB decolorization rates of 24.7% and 14.8% under visible light irradiation, respectively. The thermal ALD-grown ultra-thin-film Al_2_O_3_ layer showed an enhanced photocatalysis compared to the plasma-grown Al_2_O_3_ counterpart. The fact that the thermal ALD has a more ordered structure and narrow band gap energy boosted optical absorption in the UV and visible region and promoted the interaction. Therefore, the photocatalytic reaction would make considerably better use of the incident light and photogenerated electrons and holes, increasing the quantum efficiency. The suppressed photocatalytic activity of the combined ultra-thin-film Al_2_O_3_ layers as compared to the bare IO might be associated with the decreased electron tunnelling effect that resulted from the thickness of the Al_2_O_3_ layers [65].

## 4. Conclusions

The single TiO_2_, TiO_2_/Al_2_O_3_ thermal, and TiO_2_/Al_2_O_3_ plasma-assisted ALD-grown double-layered IO composites were successfully synthesized in the presence of a 600 nm PS opal template and microscope glass substrate. Thermal analysis maintained the adequate annealing temperature, 500 °C, which was appropriate to remove the opal completely. A well-ordered periodic microporous opal and IO structures display an FCC orientation, which is exactly planar to the surfaces of the template. A significant amount of TiO_2_ and Al_2_O_3_ were obtained in the EDX analysis due to the adequate amounts of Ti, O, and Al in the samples. The XRD and Raman analysis of the samples showed a tetragonal anatase crystal arrangement. This could be due to the inadequacy of Al_2_O_3_, which is 5 nm in amount and could be too small for XRD or Raman, even though the crystalline peaks associated with the double layers of Al_2_O_3_ were not found. The PL studies confirmed broad peaks for all samples at ~520 (2.38 eV) nm corresponding to green emissions and at 643 (3.09 eV) nm corresponding to red emissions for TiO_2_/Al_2_O_3_ thermal ALD. Therefore, the double layers have shown an interfacial charge interaction of photoexcited electron–hole pairs to restrict recombination between the layers. The UV Vis results showed a strong absorption band edge in the UV and visible regions due to the narrow energy band gaps of 3.2 eV, 3.6 eV, and 3.8 eV for TiO_2_, TiO_2_/Al_2_O_3_ thermal, and TiO_2_/Al_2_O_3_ plasma-assisted ALD-grown IO samples, respectively. It can be inferred that the lower bang gap energy and more ordered structure of thermal ALD-grown Al_2_O_3_ exhibited enhanced photocatalytic activity compared to the plasma-assisted ultra-thin-film Al_2_O_3_ composite sample.

## Figures and Tables

**Figure 1 nanomaterials-13-01314-f001:**
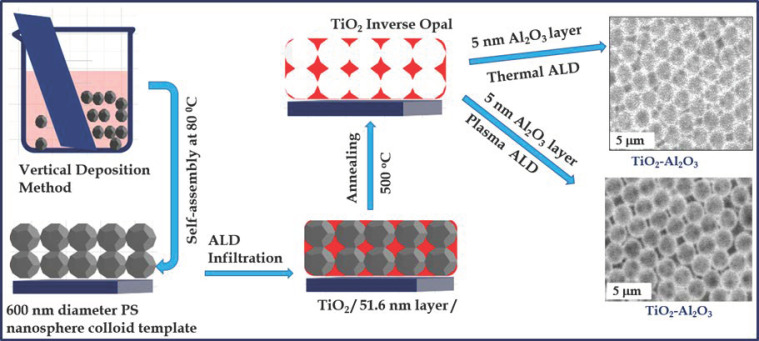
Experimental route for synthesis of the TiO_2_ IO and Al_2_O_3_/TiO_2_ composites by ALD and vertical layer deposition.

**Figure 2 nanomaterials-13-01314-f002:**
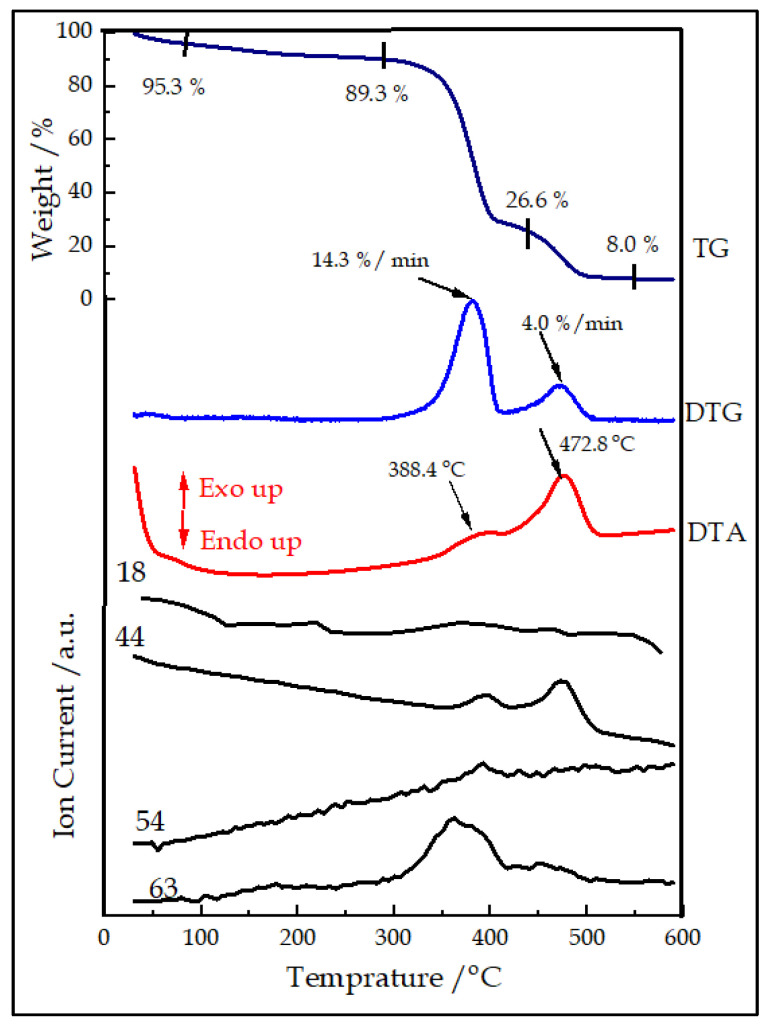
Illustrates the thermal analysis curve for PS opal template in air.

**Figure 3 nanomaterials-13-01314-f003:**
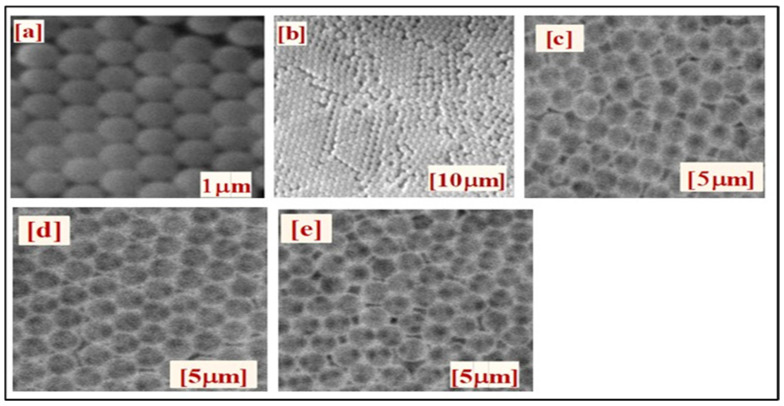
SEM images of PS opal: (**a**,**b**); pure TiO_2_ IO: (**c**); Al_2_O_3_ thermal ALD on IO: (**d**); and Al_2_O_3_ plasma ALD on IO: (**e**).

**Figure 4 nanomaterials-13-01314-f004:**
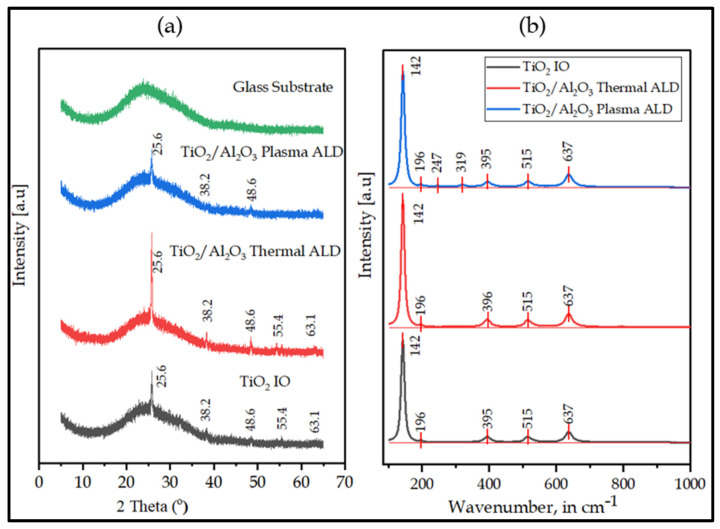
(**a**) XRD; (**b**) Raman Spectroscopy.

**Figure 5 nanomaterials-13-01314-f005:**
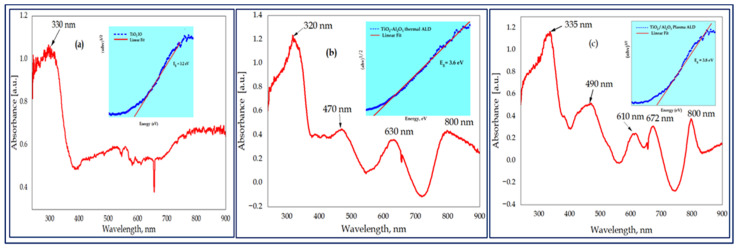
A linear fit to the straight part of the hν vs (αhν)^1/2^ plot (Tauc plot) intercepts the energy axis at the optical band gap energy: the indirect energy band gap for TiO_2_ IO (**a**), Al_2_O_3_/TiO_2_ thermal ALD, and (**b**) Al_2_O_3_/TiO_2_ plasma ALD (**c**).

**Figure 6 nanomaterials-13-01314-f006:**
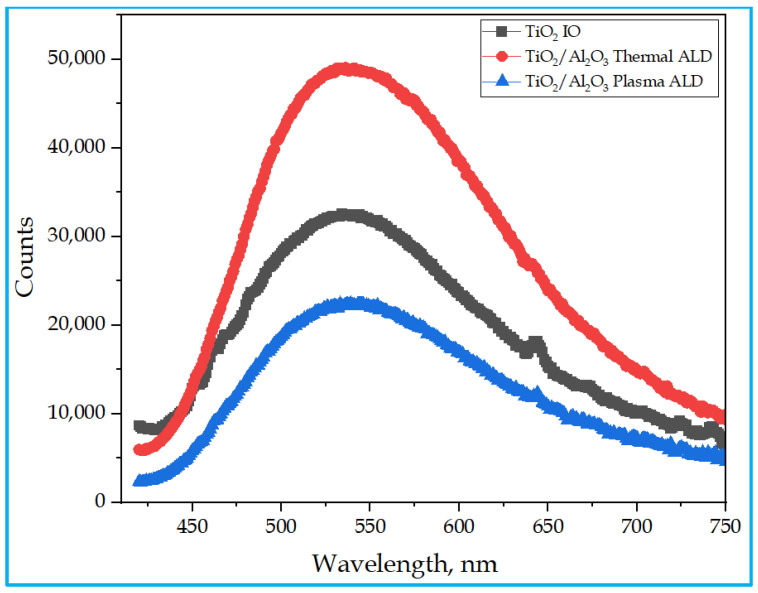
PL spectra of the samples.

**Figure 7 nanomaterials-13-01314-f007:**
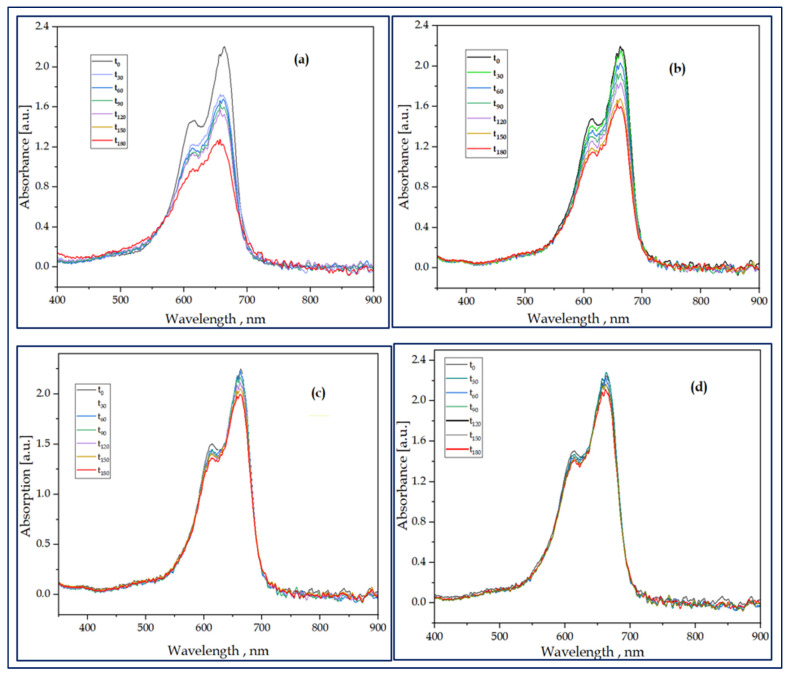
Photocatalytic decolorization of the MB dye in the presence of TiO_2_ (**a**), TiO_2_/Al_2_O_3_ thermal ALD (**b**), TiO_2_/Al_2_O_3_ plasma ALD (**c**), and MB dye (**d**).

**Figure 8 nanomaterials-13-01314-f008:**
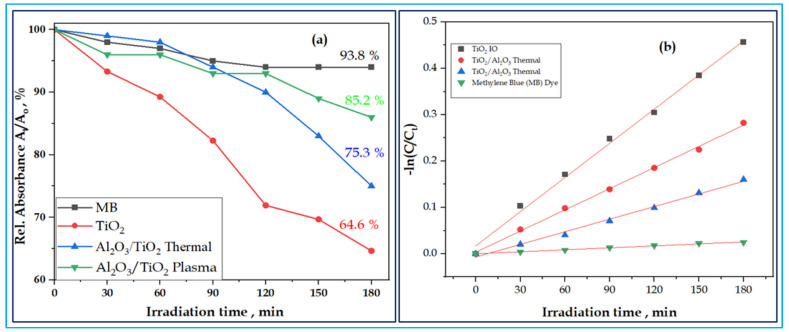
Photocatalytic performance—relative absorbance in % vs. irradiation time in minutes (**a**) and pseudo-first-order linear plots of ln (C_0_/C_t_) vs. irradiation time for the kinetics of the sample materials (**b**).

**Table 1 nanomaterials-13-01314-t001:** TiO_2_ and Al_2_O_3_ thermal and plasma-assisted ALD deposition condition.

Sample	Deposited Oxides	Temp. °C	No. of ALD Cycles	Pulse/Purge Time (s) for 1 Cycle	Thickness (nm)	Pressure (mbar)	
						(Chamber)	(Reactor)
TiO_2_	TiO_2_	52.9	671	0.3 s TiCl_4_-3 s N_2_/0.3 s H_2_O-3 s N_2_	51.6	6.8	1.3
TiO_2_/Al_2_O_3_ Thermal	Al_2_O_3_	52.9	7	0.15 s TMA-0.5 s N_2_/0.15 s H_2_O-0.5 s N_2_	5	6.8	1.4
TiO_2_/Al_2_O_3_ Plasma	Al_2_O_3_	52.9	9	0.15 s TMA-2 s N_2_/2 s O_2_-N_2_ Plasma/2 s N_2_	5	7.0	1.4

**Table 2 nanomaterials-13-01314-t002:** EDX Compositional analysis of the samples.

Sample	% *w*/*w* Ti	% *w*/*w* Al	% *w*/*w* O	% *w*/*w* C	% *w*/*w* Si	% *w*/*w* Ca	% *w*/*w* Cl	% *w*/*w* Na
TiO_2_	23.4	-	64.0	3.2	8.0	-	-	1.4
TiO_2_/Al_2_O_3_ Thermal	18.1	2.6	64.3	3.8	7.9	1.4	0.3	1.6
TiO_2_/Al_2_O_3_ Plasma	19.0	2.5	65.6	2.8	7.8	1.4	0.3	1.6

**Table 3 nanomaterials-13-01314-t003:** The rate constant, K, and linear regression square, R^2^ of the sample.

Sample	R^2^	Rate Constant, K (×10^−2^ min^−1^)
TiO_2_	0.9957	0.24
TiO_2_/Al_2_O_3_ Thermal	0.9981	0.16
TiO_2_/Al_2_O_3_ Plasma	0.9932	0.089
MB	0.994	0.014

## Data Availability

Not applicable.

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
