# Peer review of "Synthesis of TiO2/Al2O3 Double-Layer Inverse Opal by Thermal and Plasma-Assisted Atomic Layer Deposition for Photocatalytic Applications"

_nanomaterials, 2023, doi:10.3390/nano13081314_

Round 1

Reviewer 1 Report

In this study, TiO2 inverse opals (IO) and ultra-thin films of Al2O3 on IO were successfully deposited using thermal or plasma-assisted atomic layer deposition (ALD) and vertical layer deposition from a polystyrene (PS) opal template.

Strong absorption bands were found in the UV regions, including increased absorption due to slow photons and a narrow optical band gap in the visible region. The authors showed that ultrathin amorphous ALD grown Al2O3 layers have considerable photocatalytic activity. The Al2O3 thin film grown by thermal ALD has a more ordered structure compared to the plasma ALD prepared one, which explains its higher photocatalytic activity.

This is an interesting work; Nevertheless some revision are needed in order to publish this manuscript.

1. The authors could stress further the novelty of their work. As they state in the introduction part of their manuscript there are a lot of papers regarding IO TiO2.

2. Could the authors comment a little more the TG/DTG curves?

3. Regarding the photocatalytic activity of the proposed samples, the author study the decolorization of MB via Uv-Vis spectroscopy, not its degradation. UV-Vis spectroscopy cannot identy chemical bonds of MB, only its color. The authors should rephrase their findings; use the term decolorization, not degradation or decomposition. It's not correct.

4. Can the authors calculate the kinetics apparent rate for the decolorization of their samples? Does it follow a 1st order kinetics?

5. What is the incident light intensity of the light source that the authors used in mW/cm2?

6. Can the authors comment on the re-usability of their samples? Could they re-use them for 3-5 cycles and test this property? this is quite essential for real life applications..

Reviewer 2 Report

In this work, the authors prepared TiO2 and TiO2/Al2O3 inverse opals nanostructures by thermal and plasma ALD techniques. This paper is recommended for reconsideration by Nanomaterials after major revision. Please find below the specific comments.

- The purpose of this work seems confusing, which is the biggest problem of this paper. I assume that the authors aimed at improving the photocatalytic activity of TiO2 by constructing heterojunction Al2O3 surface layer, but according to the results, the constructed TiO2/Al2O3 photocatalysts showed greatly declined photocatalytic activity (Fig. 6).

- In previous reports on ALD Al2O3 surface insulating layer, the most important point is to achieve ultrathinin order to improve the performance of TiO2, otherwise the electron tunneling effect cannot happen. Obviously, the thickness of Al2O3 layers in this work are too thick, which is the reason why TiO2/Al2O3 showed declined performance than pristine TiO2. Ref: Journal of Physical Chemistry Letters 2016, 7, 1173-1179.

- Based on the above reconsideration, the Abstract and Introduction section should be revised, with the research purpose and innovation of this work being emphasized.

- Experimental evidence should be added to prove the “acceleration of separation efficiency” of photogenerated charge carriers, either by PL spectra or photocurrent density. Ref: Nanomaterials 2022, 12, 904.

- EIS results should be added to reveal the interfacial charge transfer resistance change among different samples. Ref: Nanomaterials 2021, 11, 437; etc. 

- I don’t understand why some of the chemical formulas (TiO2, Al2O3, and TiO2/Al2O3) are italic while others are regular in this manuscript.

Round 2

Reviewer 1 Report

The authors have revised their manuscript following most of the reviewers' comments.

This manuscript could be published in its present form

Reviewer 2 Report

Since the authors have addressed all the comments, I think it can be accepted by Nanomaterials in present form.